# TFCounter: Polishing Gems for Training-Free Object Counting

## Abstract

Object counting is a challenging task with broad applications in security surveillance, traffic management, and disease diagnosis. Existing methods face three major challenges: achieving superior performance, maintaining high generalizability, and minimizing annotation costs. We introduce TFCounter, a novel training-free, segmentation-based, class-agnostic object counter supporting both few-shot and zero-shot counting. This approach employs an iterative counting framework with a dual prompt system for broader recall and features a background-enhanced similarity module to improve accuracy by incorporating background context. To demonstrate cross-domain generalizability, we collected a new dataset named BIKE-1000, consisting of 1000 images of shared bicycles from Meituan. Extensive experiments on FSC-147, CARPK, and BIKE-1000 datasets show that TFCounter outperforms existing leading training-free methods and delivers competitive results compared to trained counterparts. Our code is available at https://github.com/tfcounter/TFCounter

## 1 Introduction

Object counting, the task of estimating the number of specific objects within an image, plays a crucial role in various domains, including crowd countingLiu et al. (2023a); Liang et al. (2023); Abousamra et al. (2021); Yang et al. (2022); Wang et al. (2020); Zhang et al. (2016); Peng et al. (2018); Lian et al. (2019); Sindagi et al. (2019); Zhang et al. (2015) for urban planning and security, vehicle countingHsieh et al. (2017); Mundhenk et al. (2016) for traffic management, and cell countingTyagi et al. (2023); Wang (2023); Arteta et al. (2016); Xie et al. (2018) in medical applications.

Traditional object-counting approaches are class-specific, counting objects belonging to predefined categories such as humans, cars, or cells. Typically grounded in CNN architectures, these methods require extensively annotated datasets. While exhibiting remarkable accuracy in dealing with trained categories, these methods fail to maintain their performance when counting novel classes during testing. To address this limitation, recent researchesRanjan & Nguyen (2022); Shi et al. (2022); Yang et al. (2021); Ranjan et al. (2021); ukić et al. (2023); Lu et al. (2019); Huang et al. (2024); Kang et al. (2024); Pelhan et al. (2024) have shifted towards class-agnostic object counting. They usually extract features from chosen exemplars and the query image to create a similarity map, which generates a density map to infer object count. This methodology , exemplified in ukić et al. (2023), allows for dynamic adaptation to arbitrary object classes, significantly broadening the scope and utility of object counting in computer vision.

Recent progress in class-agnostic object counting have been primarily channeled through three main axes: the training-based versus training-free axis, the density-based versus detection/segmentation-based axis, and the few-shot versus zero-shot axis (also referred to as the visual exemplar versus text exemplar axis). The former typically offers greater versatility and universality, while the latter tends to be more accurate. Current research, based on these developmental directions, aims to achieve comprehensive counting results that strike a balance between universality and precision. (i) Most of the current approaches are training-based and relies on density maps, as exemplified by Ranjan et al. (2021); Shi et al. (2022); Yang et al. (2021); You et al. (2023); ukić et al. (2023) . These methods treat the counting problem as a simple regression task, focusing more on the count values rather than precisely matching target objects, thereby reducing task difficulty. Meanwhile, end-to-end training methods often yield better precision. However, a downside is their dependence on

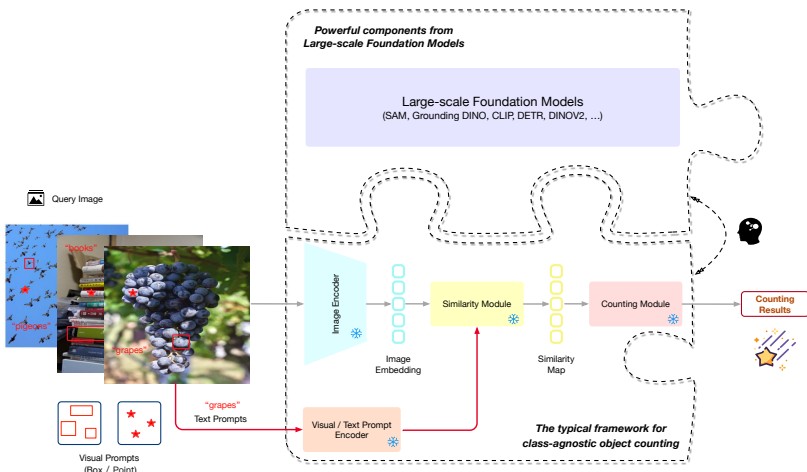

Figure 1: Integrating task-specific frameworks with generalizable components from large-scale foundation models can achieve training-free class-agnostic object counting by detailed structural design.

densely annotated counting datasets, including object points during training and bounding boxes during testing. (ii) Simultaneously, certain zero-shot modelsKang et al. (2023); Xu et al. (2023); Amini-Naieni et al. (2023); Jiang et al. (2023) utilize text prompts to identify object categories or count repeating classes in images, thus circumventing the requirement for box annotations in the testing phase. (iii) Some modelsNguyen et al. (2022); Huang et al. (2024); Shi et al. (2023) prioritize downstream versatility, not only counting the number of target objects but also generating masks or detection boxes for the targets, which provides better interpretability. However, they often lose some precision, especially when dealing with images containing dense object clusters and frequent occlusions. (iv) Additionally, the rapid advancement in large-scale foundation modelsKirillov et al. (2023); Liu et al. (2023b); Oquab et al. (2023); Radford et al. (2021); Carion et al. (2020), renowned for exceptional zero-shot generalization capabilities and flexibility in secondary development, has boosted interest in training-free approaches. Leveraging these foundation models, some methodsShi et al. (2023); Liu et al. (2023b) can perform training-free object counting by directly processing the output results or innovative structural designs, as shown in 1. Nevertheless, these methods often trade-off between high performance and broad generalizability.

In this work, focusing on greater practicality, we introduce TFCounter, as shown in Figure 2, a novel training-free, segmentation-based, class-agnostic object counter that supports both few-shot and zero-shot counting. This approach performs a multi-round counting strategy that utilizes posterior knowledge to broaden the recall scope. Subsequently, it introduces an innovative background-enhanced similarity module incorporating background context to augment accuracy. Moreover, it uses two types of points prompts, grid points prompts and residual points prompts, with the latter specifically designed to capture small objects that are often missed. This dual prompt system ensures comprehensive object detection across various sizes. Finally, to validate the effectiveness and generalizability of TFCounter, we introduce an exclusive dataset named BIKE-1000, comprising 1000 images of shared bicycles from Meituan. Experimental results show that TFCounter outperforms existing state-of-the-art training-free models on two standard counting benchmarks, and displays competitive performance when compared with training-based models. In short, our contributions can be summarized as follows:

- We introduce TFCounter, a novel training-free, segmentation-based, class-agnostic object counter which counts objects by integrating detailed structural designs with the superior advantages of large-scale foundational models.

- We propose a background-enhanced similarity module for improved precision and an iterative counting framework with a dual prompt system for broader recall.

- We present a novel exclusive dataset named BIKE-1000 for object counting, which validates the superior performance of TFCounter.

## 2 RELATED WORKS

**Zero-shot and training-free object counting.** Minimizing labor annotations was a focal point in the task of class-agnostic object counting. Existing methods frequently depended on annotations such as points and boxes during training and testing. To improve flexibility, several approaches aimed to eliminate bounding boxes during testing for zero-shot counting. Among these, EF-CACRanjan & Nguyen (2022) counted all repeating objects through the region proposal network, while ZSCXu et al. (2023), CounTXAmini-Naieni et al. (2023), CLIP-CountJiang et al. (2023), VLCounterKang et al. (2023) and PseCoHuang et al. (2024) accepted an arbitrary object class description to predict the object number. Concurrently, other methods were designed for training-free object counting, capitalizing on the robustness and generalizability inherent in large-scale foundational models. SAMKirillov et al. (2023) could perform zero-shot segmentation and subsequently estimated the number of objects by tallying all the generated masks. Based on it, SAM-FreeShi et al. (2023) combined three distinct types of class-specific priors to improve efficiency and accuracy. GroundingDINOLiu et al. (2023b) excelled in open-set detection, counting objects by aggregating detected bounding boxes. Nevertheless, zero-shot models often necessitated extensive point annotations during the training phase. Training-free methods typically struggled in complex scenes or exhibited constraints in their ability to generalize across multiple object categories.

**Improving the quality of similarity maps.** Most of class-agnostic object counting methods strived to generate high-quality similarity maps between visual features of input and example images to guide the object counting. FamNet+Ranjan et al. (2021) introduced a novel adaptation strategy for few-shot regression counting, adapting the model to new visual categories at test time with a few exemplars. BMNetShi et al. (2022) and its extension, BMNet+Shi et al. (2022), focused on a similarity-aware framework with a learnable bilinear similarity metric. CFOCNet+Yang et al. (2021) used a two-stream Resnet for different scales similarity calculation and aggregation. SAFE-CountYou et al. (2023) proposed a learning block with a similarity comparison module and a feature enhancement module, while LOCAukić et al. (2023) developed an object prototype extraction module for low-shot counting problems. However, these methods often overlooked background considerations in favor of foreground focus.

## 3 THE PROPOSED APPROACH

### 3.1 PROBLEM FORMULATION AND FRAMEWORK

In this paper, we study the challenging problem of how to enhance counting accuracy and the capability for cross-dataset generalization while adhering to the constraint of remaining training-free.

As illustrated in Figure 2, let $\mathbf{I} \in \mathbb{R}^{H \times W \times 3}$ be the input image, and let $\mathbf{B}^E = \{b_i\}_{i=1:k}$ be a set of $k$ exemplar bounding boxes denoting object exemplars. TFCounter is required to report the masks $\mathbf{M}^O = \{m_i\}_{i=1:N_O}$ of all segmented target objects along with their count.

Specifically, we introduce a novel framework named TFCounter, designed for generalized object counting and segmentation. Initially, TFCounter generates an image embedding and a mask list, followed by the production of foreground and background similarity maps via the background-enhanced similarity module. Subsequently, the prompt-aware counting module generates two types of point prompts, which are then fed into the mask decoder to generate a set of masks. An iterative counting mechanism is employed to enhance recall. Both modules are built upon the image embedding and the three key components from SAM. We detail their designs in the following sections.

### 3.2 BACKGROUND-ENHANCED SIMILARITY MODULE

We denote $f_{\theta_{\text{image}}}$, $f_{\theta_{\text{prompt}}}$, and $f_{\psi_{\text{mask}}}$ to represent the image encoder, prompt encoder, and mask decoder from SAM, respectively. Initially, we use these components to generate $k$ foreground masks $\mathbf{M}^F = \{m_i\}_{i=1:k}$ and the image embedding $\mathbf{f}^{\mathbf{I}} \in \mathbb{R}^{h \times w \times d}$.

$$\left(\mathbf{M}^F, \mathbf{f}^{\mathbf{I}}\right) = f_{\psi_{\text{mask}}}\left(f_{\theta_{\text{image}}}(\mathbf{I}), f_{\theta_{\text{prompt}}}(\mathbf{B}^E)\right) \tag{1}$$

The features of the object exemplars are extracted from $\mathbf{f}^{\mathbf{I}}$ by performing element-wise multiplication between the foreground masks and the image feature, denoted as $\mathbf{f}^{\mathbf{b}} = \mathbf{f}^{\mathbf{I}} \odot \mathbf{M}^F$, where $\odot$

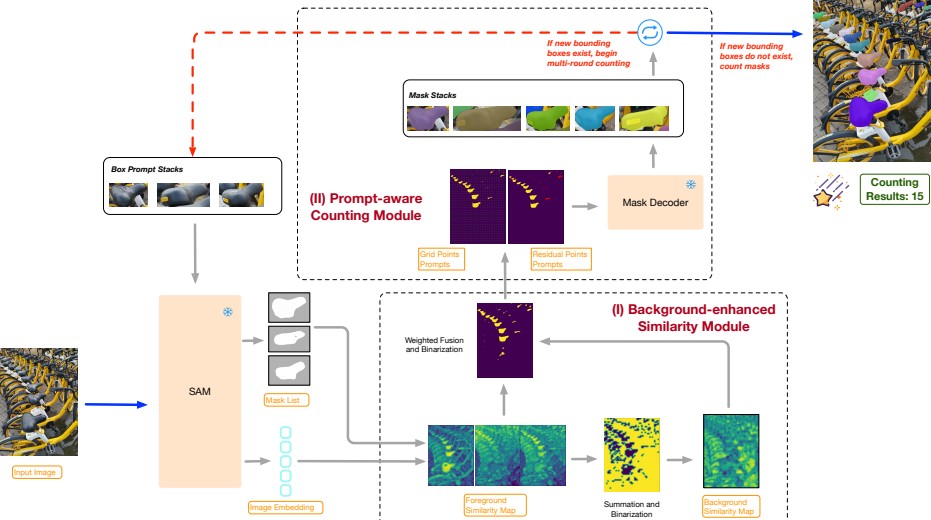

Figure 2: Overview of our TFCounter. TFCounter is a segmentation-based model designed for training-free, class-agnostic object counting. It employs an iterative counting mechanism and links two key modules: background-enhanced similarity computation, and prompt-aware object counting.

signifies the Hadamard product. The foreground similarity maps $\mathbf{Sim}^F = \left\{ sim_i^F \right\}_{i=1:k}$ between the image feature $\mathbf{f^I}$ and the exemplar feature $\mathbf{f^b}$ are computed using the cosine similarity metric.

All of the foreground similarity maps are then summed and Otsu's binarization approach Otsu et al. (1975) is applied to the result, creating a binary similarity map that serves as the background mask $\mathbf{M}^B$. Using the same method described above, we can obtain the background feature and the background similarity maps $\mathbf{Sim}^B$.

Subsequently, we assign weights to and fuse all foreground and background similarity maps. This fusion enhances the distinction between foreground and background regions for more accurate segmentation. We then apply Otsu's binarization technique once more, generating a binary composite similarity map that serves as the label map denoted as $S$.

$$S = \mathbf{T}\left( \mu + \lambda \times \mathbf{Sim}^B \right) \tag{2}$$

where $\mathbf{T}$ denotes Otsu's binarization, $\mu$ represents the average of $\mathbf{Sim}^F$, and $\lambda$ is a hyperparameter.

### 3.3 PROMPT-AWARE COUNTING MODULE

Given the label map generated in Section 3.2, the objective of this section is to provide two types of point prompts to generate the target masks.

Initially, we utilize regular $n \times n$ grid point prompts, where points where the label map is 1 are classified as positive points, and the remaining are marked as negative points, denoted as $\mathbf{P}^G$. These points are divided into batches and input into the prompt encoder and mask decoder. All the generated masks are then stored in the mask stacks $\mathbf{M}^O$.

$$f_{\psi_{\text{mask}}}\left( \mathbf{f^I}, f_{\theta_{\text{prompt}}}(\mathbf{P}^G) \right) \Rightarrow \mathbf{M}^O \tag{3}$$

Subsequently, we compare the mask stacks with the label map, labelling unmasked foreground areas where the label map is 1 as positive points, which serve as residual point prompts denoted as $\mathbf{P}^R$. The same process, as described above, is applied, primarily targeting small objects that may be missed by the grid point prompts.

$$f_{\psi_{\text{mask}}}\left( \mathbf{f^I}, f_{\theta_{\text{prompt}}}(\mathbf{P}^R) \right) \Rightarrow \mathbf{M}^O \tag{4}$$

Finally, the minimum bounding boxes generated from the mask stacks are compared with those from the prompt stacks to determine the iterative counting, which is initiated upon the detection of new bounding boxes, and continues until a predetermined iteration limit is reached.

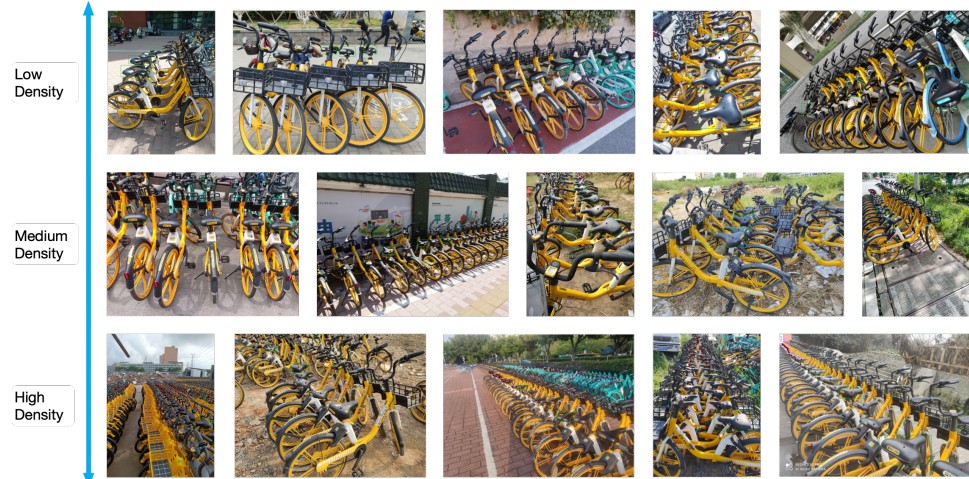

Figure 3: Few annotated images from BIKE-1000. Dot and box annotations are indicated in red and green, respectively.

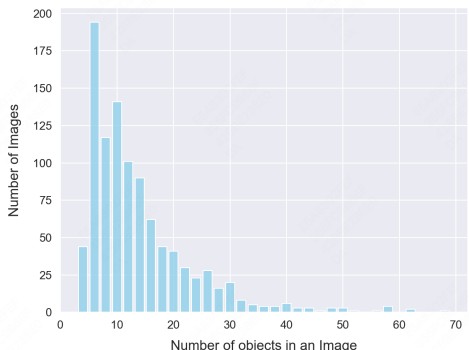

Figure 4: Number of images in several ranges of object count.

Table 1: Comparison with popular object counting datasets: "$v$" for vertical perspective, "$o$" for oblique; "$b$" for bounding box annotations, and "$p$" for point.

| Dataset | CARPK | FSC147 | BIKE-1000 |
|---|---|---|---|
| Year | 2017 | 2021 | 2024 |
| Images | 1448 | 6135 | 1000 |
| Categories | 1 | 147 | 1 |
| Instances | 43 | 56 | 13 |
| Perspective | $v$ | $v.o$ | $o$ |
| Annotation | $b.p$ | $b.p$ | $b.p$ |

### 3.4 DISCUSS

Based on SAM, SAM-Free integrates three types of priors to achieve training-free, segmentation-based, class-agnostic object counting. However, because it generates the masks only from grid points and the similarity map relies solely on input exemplars, SAM-Free performs poorly in counting small objects and is prone to misidentifying the background.

TFCounter draws inspiration from SAM-Free and adheres to a similar framework. To address these issues, we utilize background context to enhance the discriminative potency of the similarity map. Additionally, we employ an iterative counting framework with a dual prompt system to achieve broader recall. Detailed qualitative and quantitative comparisons are provided in Section 5.2.

## 4 BIKE-1000 DATASET

This paper utilizes exclusive data from Meituan, one of China's leading shared bicycle enterprises. In the bike-sharing and ebike-sharing industry, accurate bicycle counting is a central requirement across multiple application scenarios, including orderly operations management, inventory audits, and street silt removal. To support these scenarios and advance the research and development of more precise and efficient counting technologies, we have established a novel object counting dataset named BIKE-1000. This dataset provides a large collection of bicycle images accompanied by their

count annotations, which aids in improving bicycle management, enhancing operational efficiency, and ultimately optimizing the user experience.

The BIKE-1000 dataset encompasses a collection of 1000 images, each featuring distinctly visible shared bicycles situated within various scenes. These images were primarily captured by operators. A significant portion of the dataset is characterized by photographs taken from an oblique perspective, which presents the bicycles with considerable variations in shape, appearance, and size, as well as instances of partial occlusion. Such attributes pose typical challenges in the domain of object counting in computer vision. The annotation protocol for the BIKE-1000 dataset adheres to the methodology used in FSC147 Ranjan et al. (2021), comprising (1) point annotation, where each countable bicycle seat is marked, and (2) bounding box annotation, with three instances per image demarcated as examples. The dataset includes high-resolution imagery with bicycles ranging from 3 to 70 per image, averaging 13 objects. Note that shared bicycles consist of numerous components, such as frames, handlebars, wheels, seats, etc., whose appearance can vary significantly when viewed from different angles. Manually counting over 70 bicycle seats in a single image proved difficult, especially in images with oblique perspective. Therefore, we have limited our image selection to those with fewer than 70 bicycle seats for the BIKE-1000 dataset. The visualizations are displayed in Figure 3, while the statistical data and comparisons with object count benchmarks are shown in Figure 4 and Table 1.

## 5 EXPERIMENTS

### 5.1 EXPERIMENTAL SETUP

**Dataset.** We evaluate TFCounter on two general object counting datasets, FSC147 and CARPK, and further study its generalizability on the proposed BIKE-1000. FSC147 contains 6135 images spanning 147 object categories, with a test subset of 1190 images from 29 categories. CARPK includes 1448 images documenting around 90,000 cars from a drone's perspective, with 459 images dedicated to testing. The BIKE-1000 dataset, with its complete set of 1000 images, serves to estimate model's performance in a novel domain.

**Evaluation metrics.** We report the Mean Absolute Error (MAE), Root Mean Square Error (RMSE), Normalized Relative Error (NAE), and Squared Relative Error (SRE) metrics. These metrics are defined as follows: $\text{MAE} = \frac{1}{n} \sum_{i=1}^{n} |y_i - \hat{y}_i|$, $\text{RMSE} = \sqrt{\frac{1}{n} \sum_{i=1}^{n} (y_i - \hat{y}_i)^2}$, $\text{NAE} = \frac{1}{n} \sum_{i=1}^{n} \frac{|y_i - \hat{y}_i|}{y_i}$ and $\text{SRE} = \sqrt{\frac{1}{n} \sum_{i=1}^{n} \frac{(y_i - \hat{y}_i)^2}{y_i}}$, where $n$ denotes the number of test images, and $y_i$ and $\hat{y}_i$ represent the actual and predicted object counts, respectively.

**Implementation details.** In the weighted fusion process of foreground and background similarity maps, we adjust $\lambda$ to 0.5 for FSC147 and to 0.7 for CARPK and BIKE-1000. Moreover, to prevent small objects from being omitted by excessive background fusion, $\lambda$ is set to 0 when the foreground regions are more than 50%.

### 5.2 STATE-OF-THE-ART COMPARISON

We compare our model to competitive baselines: (1) CFOCNetYang et al. (2021), (2) FamNetRanjan et al. (2021), (3) BMNet+Shi et al. (2022), (4) CounTRLiu et al. (2022), (5) LOCAukić et al. (2023), (6) CACViTWang et al. (2024), (7) COUNTGDAmini-Naieni et al. (2024), (8) DAVEPelhan et al. (2024), (9) Counting-DETRNguyen et al. (2022), (10) PseCoHuang et al. (2024), (11) SAM-FreeShi et al. (2023), (12) ZSCXu et al. (2023), (13) CLIP-CountJiang et al. (2023), (14) VLCounterKang et al. (2024), (15) CounTXAmini-Naieni et al. (2023), (16) SAMKirillov et al. (2023), (17) GroundingDINOLiu et al. (2023b).

**Results on Few-shot/One-shot Object Counting.** In the few-shot counting scenario, each image provides three bounding box annotations of exemplar objects, which are used to count the target objects in the image. Table 2 offers quantitative comparisons with recent state-of-the-art methods, including detection/segmentation-based versus density-based approaches, and training-based versus training-free approaches. Our TFCounter exhibits significant advancements over existing training-free methodologies, irrespective of the utilization of point prompts or bounding box

Table 2: Few-shot object counting on FSC147 and CARPK. The best performance in each group is highlighted in bold.

| Methods | Venue | Prompt | Output | FSC147 | | CARPK | |
|---|---|---|---|---|---|---|---|
| | | | | MAE | RMSE | MAE | RMSE |
| *Training-based* | | | | | | | |
| CFOCNet | WACV'21 | box | density | 22.10 | 112.71 | - | - |
| FamNet | CVPR'21 | box | density | 22.08 | 99.54 | 28.84 | 44.47 |
| BMNet+ | CVPR'22 | box | density | 14.62 | 91.83 | 10.44 | 13.77 |
| CounTR | BMVC'22 | box | density | 11.95 | 91.23 | - | - |
| LOCA | ICCV'23 | box | density | 10.79 | 56.97 | 9.97 | 12.51 |
| CACViT | AAAI'24 | box | density | 9.13 | 48.96 | **8.30** | **11.18** |
| COUNTGD | ArXiv'24 | box | density | **8.31** | 91.05 | - | - |
| DAVE | CVPR'24 | box | density&detection | 8.66 | **32.36** | - | - |
| Counting-DETR | ECCV'22 | box | detection | 16.79 | 123.56 | - | - |
| PseCo | CVPR'24 | box | detection | **13.05** | **112.86** | - | - |
| *Training-free* | | | | | | | |
| SAM-Free | WACV'24 | point | segmention | 20.10 | 132.83 | 11.01 | 14.34 |
| TFcounter | Ours | point | segmention | 18.58 | 131.99 | **8.94** | **11.56** |
| SAM-Free | WACV'24 | box | segmention | 19.95 | 132.16 | 10.97 | 14.24 |
| TFcounter | Ours | box | segmention | **18.41** | **130.50** | 9.71 | 12.44 |

Table 3: One-shot/zero-shot object counting on the FSC147.

| Scheme | Methods | Venue | Prompt | Output | FSC147 | |
|---|---|---|---|---|---|---|
| | | | | | MAE | RMSE |
| | *Training-based* | | | | | |
| | CounTR | BMVC'22 | box | density | 12.06 | 90.01 |
| | LOCA | ICCV'23 | box | density | 12.53 | 75.32 |
| | DAVE | CVPR'24 | box | density&detection | 11.29 | 66.36 |
| One-shot | CACViT | AAAI'24 | box | density | **8.62** | **29.92** |
| | *Training-free* | | | | | |
| | SAM-Free | WACV'24 | point | segmention | 21.83 | 136.77 |
| | TFcounter | Ours | point | segmention | **19.86** | 135.54 |
| | SAM-Free | WACV'24 | box | segmention | 21.60 | 136.36 |
| | TFcounter | Ours | box | segmention | 19.88 | **135.09** |
| | *Training-based* | | | | | |
| | ZSC | CVPR'23 | text | density | 22.09 | 115.17 |
| | CLIP-Count | ACM MM'23 | text | density | 17.78 | 106.62 |
| | VLCounter | AAAI'24 | text | density | 17.05 | 106.16 |
| | CounTX | BMVC'23 | text | density | 15.88 | 106.29 |
| | DAVE | CVPR'24 | text | density&detection | 14.90 | 103.42 |
| Zero-shot | COUNTGD | arXiv'24 | text | density | **12.98** | **98.35** |
| | PseCo | CVPR'24 | text | detection | **16.58** | **129.77** |
| | *Training-free* | | | | | |
| | SAM | ICCV'23 | None | mask | 42.48 | 137.50 |
| | GroundingDINO | arXiv'23 | text | segmention | 62.47 | 160.09 |
| | SAM-Free | WACV'24 | text | segmention | 29.16 | 137.05 |
| | TFcounter | Ours | text | segmention | **26.13** | **135.51** |

prompts. Furthermore, it attains comparable MAE and RMSE metrics to the state-of-the-art training-based detection-based methods. Similarly, in the one-shot object counting scenario, each image provides a single bounding box annotation of exemplar objects. The results, shown in Table 3, indicate that TFCounter significantly outperforms existing training-free methods.

**Results on Zero-shot Object Counting.** In the zero-shot counting scenario, each image provides a text description of the counting category. Following SAM-Free, we employ CLIP-SurgeryLi et al. (2023) to calculate the initial similarity between the image and text representations. This similarity

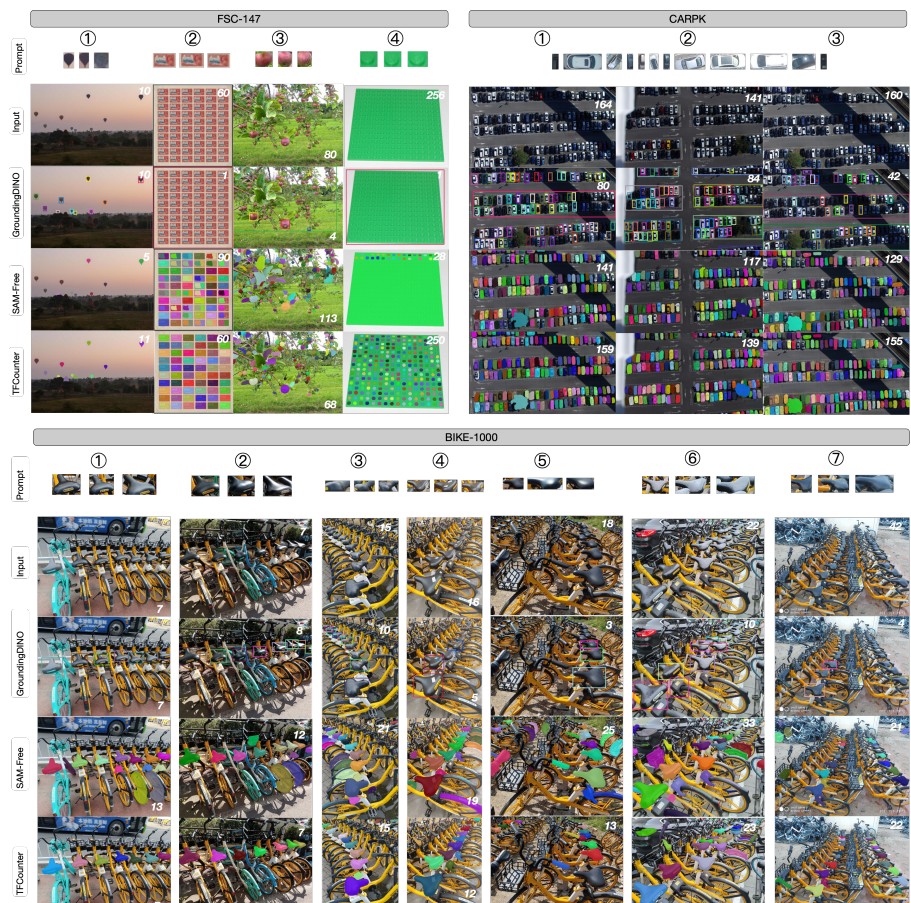

Figure 5: Qualitative comparison on FSC147, CARPK, and BIKE-1000.

is subsequently used to select exemplar objects through region selection and box creation. The results are shown in Table 3. Note that the zero-shot mode of SAM-Free shows differences in replication performance compared to the original paper. The results in this paper are from replicated experiments. The performance of TFCounter is obtained under the same settings.

**Results on Training-free Object Counting.** We compared the performance of training-free methods TFCounter and SAM-Free across three datasets. SAM-Free is currently the state-of-the-art method in the training-free, class-agnostic object counting field. The comparison results are shown in Table 4. MAE and RMSE indicate the average error per image across the dataset and are more sensitive to high-density images. NAE and SRE indicate the relative error per image and are more sensitive to inter-class variance. Due to the squared amplification effect, RMSE and SRE emphasize a few extreme errors. The results show that TFCounter outperforms SAM-Free in all metrics across the three datasets, with a significant improvement in NAE. Notably, on the BIKE-1000 dataset, both TFCounter and SAM-Free have lower MAE and RMSE compared to FSC147 and CARPK, likely due to the lower object density. However, NAE is higher due to greater average inter-class variance - caused by variations in object scale, pose, and overlap. SRE, on the other hand, is lower because the counting difficulty is more balanced, resulting in fewer extreme errors.

**Visualization.** Figure 5 illustrates the qualitative distinctions among several training-free models. GroundingDINO performs commendably well in counting low-density objects but struggles with high-density object counting and significant intra-class variations. In contrast, SAM-Free surpasses GroundingDINO in high-density scenarios but tends to produce false positives, misidentifying non-target items that resemble target objects in shape or color. For instance, in the BIKE-1000 test images, SAM-Free frequently mistakes bike locks and wheels. Furthermore, SAM-Free often fragments a single object into multiple parts, as shown in example ② in FSC147. Our TFCounter

Table 4: Training-free object counting on three datasets, while * denotes points prompts.

| Methods | FSC147 | | | | CARPK | | | | BIKE-1000 | | | |
|---|---|---|---|---|---|---|---|---|---|---|---|---|
| | MAE | RMSE | NAE | SRE | MAE | RMSE | NAE | SRE | MAE | RMSE | NAE | SRE |
| SAM-Free* | 20.10 | 132.83 | 0.30 | 3.87 | 11.01 | 14.34 | 0.51 | 3.89 | 7.65 | 10.26 | 0.73 | 2.86 |
| TFcounter* | 18.58 | 131.99 | **0.28** | 3.85 | **8.94** | **11.56** | **0.41** | **3.18** | 6.69 | 10.07 | 0.54 | 2.30 |
| SAM-Free | 19.95 | 132.16 | 0.29 | **3.80** | 10.97 | 14.24 | 0.48 | 3.70 | 7.43 | 10.07 | 0.68 | 2.66 |
| TFcounter | **18.41** | **130.50** | **0.28** | 3.84 | 9.71 | 12.44 | 0.47 | 3.67 | **6.58** | **10.00** | **0.50** | **2.18** |

Table 5: Ablation study on each component of TFCounter. The optimal and suboptimal results are represented in red and blue, respectively.

| No. | Background Similarity | Multi-round Counting | Residual Points Prompts | FSC147 | | | | BIKE-1000 | | | |
|---|---|---|---|---|---|---|---|---|---|---|---|
| | | | | MAE | RMSE | NAE | SRE | MAE | RMSE | NAE | SRE |
| M0 | ○ | ○ | ○ | 19.95 | 132.16 | 0.29 | 3.80 | 7.43 | 10.07 | 0.68 | 2.66 |
| M1 | ● | ○ | ○ | 20.85 | 132.38 | 0.24 | 3.68 | 10.55 | 14.60 | 0.66 | 3.05 |
| M2 | ○ | ● | ○ | 20.62 | 132.04 | 0.33 | 4.23 | 9.16 | 11.80 | 0.93 | 3.61 |
| M3 | ○ | ○ | ● | 21.41 | 131.94 | 0.39 | 4.50 | 22.32 | 25.74 | 2.36 | 8.06 |
| M4 | ● | ● | ○ | 20.46 | 131.79 | 0.26 | 3.75 | 10.36 | 14.43 | 0.65 | 3.01 |
| M5 | ● | ○ | ● | 18.50 | 130.92 | 0.25 | 3.73 | 6.48 | 10.40 | 0.44 | 2.08 |
| M6 | ○ | ● | ● | 23.22 | 132.25 | 0.47 | 5.06 | 30.16 | 35.31 | 3.17 | 11.35 |
| M7 | ● | ● | ● | 18.41 | 130.50 | 0.28 | 3.84 | 6.58 | 10.00 | 0.50 | 2.18 |

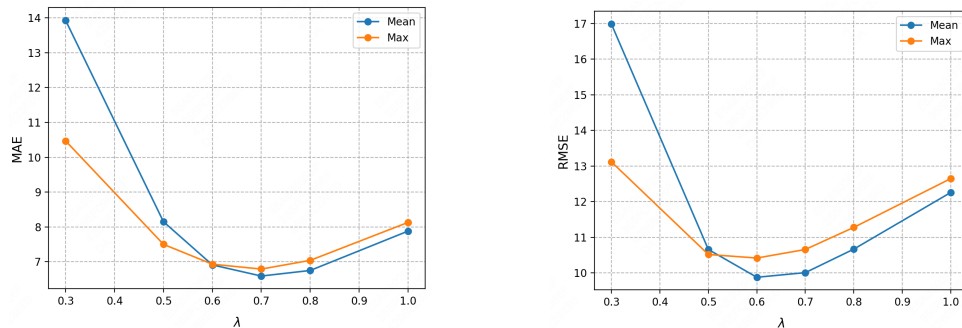

Figure 6: Influence of hyperparameter $\lambda$ in the weighted fusion of similarity maps.

notably addresses these limitations. By focusing solely on quantity rather than localization, metrics like MAE and NAE may not fully capture the improvements made by TFCounter. However, these enhancements are evident in the visual comparisons, as illustrated in example ④ in BIKE-1000.

## 5.3 ABLATION STUDIES AND ANALYSIS

**Component Analysis.** To validate the effectiveness of each component, we conduct an ablation study as presented in Table 5. Starting with SAM-Free (M0), we add Background Similarity, Multi-round Counting, and Residual Points Prompts in M1, M2, and M3, respectively, and then combine them pairwise in M4, M5, and M6. Among these experiments, the best performance is observed in M7, followed by M5, while M6 performs the worst. Additionally, the performance of each individual component such as M1, M2, and M3 is worse than M0. This occurs because each component has a unique function: Background Similarity filters out irrelevant masks to boost accuracy, while Multi-round Counting and Residual Points Prompts expand the recall scope to include more target objects. The best performance comes from combining these components, as Background Similarity alone may exclude smaller objects, and Multi-round Counting or Residual Points Prompts alone may include non-target objects.

**Hyperparameters Analysis.** We investigate the influence of the hyperparameter $\lambda$ in the weighted fusion of similarity maps. Two fusion methods are tested: 1) "Mean" fusion, formulated as $S = \mathbf{T}(\mu + \lambda \mathbf{Sim}^B)$, where $\mu$ is the mean value of $\mathbf{Sim}^F$; and 2) "Max" fusion, formulated as $S =$

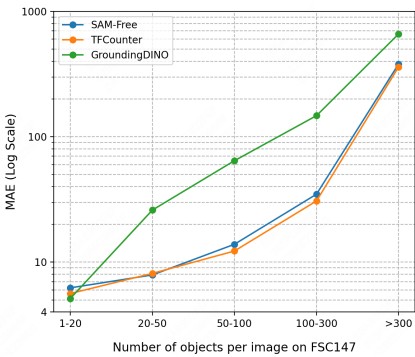 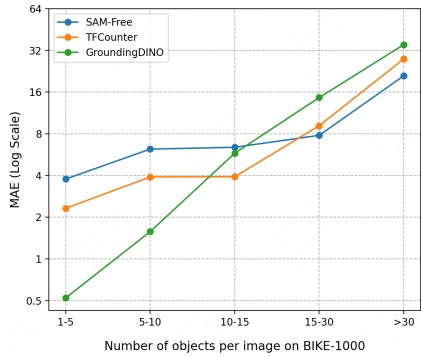

Figure 7: Performance in different density images.

$\mathbf{T}(\phi + \lambda \mathbf{Sim}^B)$, where $\phi$ is the maximum value of $\mathbf{Sim}^F$. Figure 6 shows the impact of different $\lambda$ values and fusion methods on the BIKE-1000 dataset. As $\lambda$ increases, both MAE and RMSE initially decrease and then rise, with the optimal point slightly differing between 0.7 and 0.6. This suggests an optimal fusion ratio for the BIKE-1000 dataset. Fine-tuning this ratio for each image could improve accuracy and is a potential direction for future research. In this paper, the "Mean" method with $\lambda = 0.7$ is adopted for the BIKE-1000 dataset and CARPK, while the "Mean" method with $\lambda = 0.5$ is used for FSC147.

**Density Analysis.** We compared three training-free methods on test images with varying densities. Figure 7 presents the MAE on the FSC147 and BIKE-1000 datasets, using a logarithmic scale on the vertical axis for better visualization. GroundingDINO performs best on low-density images, but its MAE increases rapidly with density. TFCounter demonstrates superior accuracy in medium to low-density scenarios. Conversely, SAM-Free outperforms TFCounter in high-density images. However, SAM-Free's performance in high-density images is partly due to recalling more non-target objects, which unexpectedly brings the counting results closer to the true value. An example of this can be seen in example ④ in the BIKE-1000 dataset of Figure 5.

## 6 LIMITATIONS

The initial version of TFCounter presents several limitations. **Segmentation Problem.** As a segmentation-based approach, TFCounter struggles with high-density overlapping objects compared to density-based methods. It also faces double-counting issues, especially when segmenting objects with multiple parts, such as the red flesh and green calyx of a strawberry. Future research on more fine-grained or improved semantic/instance segmentation models could help mitigate these limitations. **Train-free Problem.** TFCounter's train-free design avoids large annotated training datasets and reduces computational overhead. However, relying on manually designed structures limits its generalization. Future work will explore training components such as prompt selection and similarity fusion, or using techniques like LoRA to improve performance.

## 7 CONCLUSIONS

In this paper, we explore an intriguing question: how to adapt large-scale foundation models to various downstream tasks and domain data without training, while maintaining superior performance. To this end, we introduce TFCounter, a novel training-free, segmentation-based, class-agnostic object counter that supports both few-shot and zero-shot counting. The originality of TFCounter stems from three core designs: a multi-round counting strategy, a dual prompt system, and a background-enhanced similarity module. The first two broaden the recall scope, while the latter boosts accuracy by incorporating background context. Experimental results show that TFCounter outperforms existing state-of-the-art training-free models on two standard counting benchmarks and the proposed BIKE-1000, and displays competitive performance compared to training-based models. Future work will focus on improving the counting of high-density overlapping objects and developing lightweight training for better performance.

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
