# OpenReview forum: "TFCounter: Polishing Gems for Training-Free Object Counting"
_ICLR.cc/2025/Conference — ICLR 2025 Conference Withdrawn Submission_

### Official Review · Reviewer_vfqu · 2024-10-20

**Soundness:** 4
**Presentation:** 4
**Contribution:** 3
**Rating:** 6
**Confidence:** 4

**Summary:**

The paper proposes a training-free, class-agnostic object counter through a count-by-detection approach using SAM. It includes a similarity module with background context to boost precision and is prompted with course and fine point prompts. Finally, they also propose a new dataset for object counting, including 1000 images of bicycles.

**Strengths:**

I largely enjoyed reading the paper. Overall, it was very clear, well-organized, and presented motivating results in object counting. I thought that the most interesting figures were Figure 5, the qualitative comparison of datasets across TFCounter, SAM-Free, and GroundingDINO vs. the inputs, and the ablation studies. The former offered insight on model performance, while the latter showcased TFCounter's areas of strength and areas of improvement. I am admittedly not an expert in training-free approaches, but I really encourage the community to work more in this space due to its resource and computational efficiency. On that note, it would be useful, possibly in conclusions or somewhere in the discussions, to list what resources were used for this paper and how fast the experiments were to conduct.

**Weaknesses:**

Re novelty: It would be useful to include additional comparative experiments with similar and/or updated architectures. Without these comparisons, it is challenging to say if this training-free count method is SOTA.

For example:
How does similar/different is this proposed approach from CellSAM https://github.com/vanvalenlab/cellSAM?

How does the performance shift when SAM2 (released this summer) is used out of the box? How does performance shift when replacing SAM2 in the pipeline?

Re readability: The paper lists several evaluation metrics, but fails to mention why they include so many of them. It would be useful to add 1-2 sentences on why/how the metrics complement each other, and that they are commonly used, novel, etc. Similar comment on the choice of the hyperparameter lambda - 0.5 and 0.7 were chosen for different datasets - why?

**Questions:**

It would be useful to describe how the number k (k foreground masks) was decided upon.

Not a knock, more an unknown - why does the paper use the Hadamard product rather than an inner product or other type of product?

Was all of SAM frozen (i.e. SAM was just used for inference), was SAM's backbone/neck frozen? This clarification would help.

It would be useful to clarify that FSC147 147 categories do/do not include bicycles.

Suggestions:
The paper introduces terminology i.e. Sim^B, M^B, etc which would likely be useful to include in the Figure and Figure caption.

From my understanding, the residual point prompts are those not captured by the geometric grid of point prompts. It makes sense to have these, but the name residual makes me think that these point prompts reference smaller objects. It makes sense that larger objects are more likely to be captured by the grid-prompts, but it could be the case that a larger object falls between grid prompts (say, for example, the task was to count the neck of the bicycles rather than the seat). Adding some extra sentence on this might be useful for clarification on intent.

It would be useful to list results not in a table, but rather in a plot. Or include both, with the table as suplemental. The table is challenging to parse - particularly tables 4 and 5.

---

### Official Review · Reviewer_duZx · 2024-11-02

**Soundness:** 2
**Presentation:** 2
**Contribution:** 2
**Rating:** 3
**Confidence:** 4

**Summary:**

This paper introduces a training-free model for few-shot and zero-shot object counting. Building upon SAM-Free, the authors propose two modules to improve counting performance, including a background-enhanced similarity module and a prompt-aware counting module. The former is used to boost accuracy while the latter is adopted to enhance recall. Experiments on a few counting datasets show that the proposed method outperforms the baseline.

**Strengths:**

1. This paper proposes two customized modules to improve the counting performance of SAM-Free.
2. Experimental results demonstrate that the proposed approach achieves competitive results against previous methods.

**Weaknesses:**

1. There is no strong evidence to support the effectiveness of the proposed modules. Table 5 indicates that using the proposed modules individually leads to a decline in performance. While the authors argue that the background-enhanced similarity module enhances accuracy and the prompt-aware counting module improves recall, they do not provide quantitative results to support these claims.
2. The performance gain over the baseline appears marginal. For example, TFCounter achieves an MAE of 18.41 on the FSC-147 dataset, which is slightly better than the baseline (19.95 MAE).
3. Missing analysis on inference efficiency. The proposed modules involve multi-round counting during inference, which could be computationally expensive.
4. The proposed TFCounter does not consistently outperform SAM-Free. Figure 7 shows that SAM-Free outperforms TFCounter on the BIKE-1000 dataset when the number of objects exceeds 15.
5. It seems that the proposed method requires careful hyperparameter tuning. Line 305 indicates that the selection of lambda varies depending on the specific scenario.

**Questions:**

1. What is the inference speed of TFCounter?

---

### Official Review · Reviewer_k12Q · 2024-11-03

**Soundness:** 2
**Presentation:** 1
**Contribution:** 1
**Rating:** 1
**Confidence:** 5

**Summary:**

This paper discusses training-free generic object counting. This  method is training-free, segmentation-based, class-agnostic object counter that supports both few-shot and zero-shot counting. Authors also introduced a new dataset named BIKE-1000 for object counting.

**Strengths:**

The proposed system seems effective, in comparison wIth the SAM-Free counting method.

**Weaknesses:**

- This paper is not very well motivated. The authors spend a few paragraphs in the Introduction section discussing prior categories of works. These paragraphs are generic and offer limited information about the problem this paper is solving and how the proposed method solves this problem.

- The authors didn't provide a convincing statement of novelty. The design of the method looks complicated. I cannot tell from this what the novelty is and what insights are offered. This questions is strongly related to the motivation question above.

- Writing needs lots of improvement. There are numerous citation formatting issues. There should be a branket at most citation places. English should be improved too.

- The figures should be improved to look more professional.

- I would suggest not including the training-based methods in the experiment part. They are not helpful because they are of a different strategy and thus are not comparable.

- Overall, this paper is not well polished. I would suggest authors polishing the paper heavily before submiting it to a different place.

**Questions:**

See comments above.

---

### Official Review · Reviewer_85Be · 2024-11-04

**Soundness:** 3
**Presentation:** 3
**Contribution:** 2
**Rating:** 5
**Confidence:** 4

**Summary:**

In this work, the authors introduce TFCounter, a training-free, segmentation-based, class-agnostic object counter supporting both few-shot and zero-shot counting.

This approach employs an iterative counting framework with a dual prompt system for broader recall and features a background-enhanced similarity module to improve accuracy by incorporating background context.

**Strengths:**

Strengths.

  + The proposed background-enhanced similarity module and iterative counting framework are reasonable.
  +  A new dataset BIKE-1000 for object counting is collected.
  +  The experiments of the proposed method are adequate.

**Weaknesses:**

Weaknesses.

- The authors summarize the first contribution of this paper as "TFCounter counts objects by integrating detailed structural designs with the superior advantages of large-scale foundational models." However, the baseline SAM-Free (M0 in Table 5) has already applied SAM to the object counting task. It cannot be considered a contribution of this paper.

- The results of the ablation study are less ideal. The design of background similarity and the multi-round counting are not strongly correlated. The reason why the sole use of them led to performance degradation needs to be further discussed.

- The quantitative results of GroundingDINO on few-shot / one-shot object counting are absent. Only visualization results of GroundingDINO on few-shot / one-shot object counting (Figure 5) are given.

- In Section 2, the related work of few-shot object counting is absent. However, a large portion of the experiments, including ablation study in this paper, are focused on few-shot object counting.

- In Figure 2, when new bounding boxes exist, the red dotted line points to Boxes Prompt Stacks. Is it iterative from scratch? It may be sufficient to start iterating from the similarity computation. Which variables will change during each iteration?

-The paper has several ambiguities or errors in the text.

(1) In line 047, what do "the former" and "the latter" refer to?

(2) In lines 049-086, The authors list four points to discuss the current research, which is confusing and lacking in logic, e.g., the (ii) discussion of zero-shot models is not very relevant to this paper. In (iii), "some models not only count the number of target objects but also generate masks or detection boxes for the targets, which often lose some precision." The proposed method TFCounter also generates the masks, whether it also suffers from loss of precision.

(3) In line 107, "The presentation of dataset BIKE-1000 validates the superior performance of TFCounter." is irrational, which should be "the experiments on three datasets validates …"

(4) "TFCounter outperforms SAM-Free in all metrics across the three datasets, with a significant improvement in NAE." It seems that TFCounter's SAE on FSC147 is weaker, and the significant improvement in NAE only is seen on BIKE-1000 and CARPK.

(5) In line 478, "Background Similarity filters out irrelevant masks to boost accuracy." It seems that background similarity does not have this utility. It can enhance the relevant masks.

**Questions:**

See Weaknesses.

---

### Note · Authors · 2024-11-23

I have read and agree with the venue's withdrawal policy on behalf of myself and my co-authors.